# Trends in Antibiotic Use in a Large Children’s Hospital in London (United Kingdom): 5 Years of Point Prevalence Surveys

**DOI:** 10.3390/antibiotics13020172

**Published:** 2024-02-09

**Authors:** Kevin Meesters, Faye Chappell, Alicia Demirjian

**Affiliations:** 1Department of Paediatric Infectious Diseases and Immunology, Evelina London Children’s Hospital, London SE1 7EH, UK; faye.chappell@gstt.nhs.uk (F.C.); alicia.demirjian@ukhsa.gov.uk (A.D.); 2Department of Pediatrics, University of British Columbia, Vancouver, BC V6H 3V4, Canada; 3Vaccine Evaluation Center, BC Children’s Hospital Research Institute, University of British Columbia, Vancouver, BC V5Z 4H4, Canada; 4Healthcare-Associated Infection (HCAI), Fungal, Antimicrobial Resistance (AMR), Antimicrobial Use (AMU) & Sepsis Division, United Kingdom Health Security Agency (UKHSA), London NW9 5EQ, UK; 5Faculty of Life Sciences & Medicine, King’s College London, London WC2R 2LS, UK

**Keywords:** antimicrobial stewardship, point prevalence study, antibiotic utilisation

## Abstract

*Background*: Antibiotics are commonly prescribed in paediatrics. As their excessive use contributes to adverse drug events, increased healthcare costs, and antimicrobial resistance, antimicrobial stewardship initiatives are essential to optimising medical care. These single-centre point prevalence surveys aimed to provide insights into antibiotic prescribing trends and identify targets for paediatric AMS activities. *Methods*: 14 point prevalence surveys were conducted from March 2016–April 2021, collecting data on antibiotic prescriptions, indication, adherence to guidelines, and route of administration. The UK adapted the World Health Organisation’s AWaRe classification-guided antibiotic categorization. *Results*: 32.5% of all inpatients were on at least one antimicrobial; this remained stable during all surveys (range: 20–44%, *p* = 0.448). Of all prescriptions, 67.2% had an end- or review-date, and the majority was for agents in the Watch category (46.8–70.5%). Amoxicillin and clavulanate were the most frequently prescribed antibiotics (20.8%), followed by gentamicin (15.3%). Approximately 28.8% of all prescriptions were for prophylactic indications, while 7.6% of the prescriptions were not adherent to the hospital guidelines. *Conclusions*: This study highlights the importance of ongoing monitoring and robust AMS initiatives to ensure prudent antibiotic prescribing in paediatric healthcare. It underscores the need for tailored guidelines, educational efforts, and targeted interventions to enhance the quality of antibiotic usage, ultimately benefiting both individual patients and public health.

## 1. Introduction

Antibiotics are the most frequently prescribed group of medications for children in hospitals [1,2]. Unnecessary antibiotic administration not only leads to adverse effects and increased healthcare costs but also significantly contributes to the emergence of antimicrobial resistance [3]. Therefore, when prescribing empiric antibiotics, it is crucial to consider factors such as the antibiotic’s spectrum, dosage, route of delivery, and treatment duration [4]. Furthermore, prescribers need to routinely review new microbiological findings throughout the course of treatment and closely track the patient’s clinical progress. Ideally, this would lead to a more targeted treatment, including the possible discontinuation of antibiotic therapy. 

In paediatric healthcare delivery, inpatient antimicrobial stewardship (AMS) initiatives, which consist of a coordinated set of measures to monitor and enhance the quality of antibiotic usage, are increasingly prevalent [5]. However, the size and disciplines involved in paediatric AMS programmes can vary significantly [6]. Measuring antibiotic prescribing is crucial, as these data can help guide AMS interventions. However, there is no universally accepted metric for antibiotic prescribing in the paediatric inpatient setting. The World Health Organisation’s (WHO) AWaRe classification has been introduced to assess and establish targets for stewardship activities [7], with a target of at least 60% of all prescribed antibiotics, at a country-level, in the Access class by 2023. This classification was adapted by Public Health England, now the UK Health Security Agency (UKHSA), for use in the United Kingdom (Appendix A) [8]. Antibiotics in the Access category are considered first-line agents for common infections since they are safe, effective, and less likely to develop resistance. In contrast, the Watch category includes antibiotics with a higher potential for resistance, while those in the Reserve category are regarded as last-resort options.

The primary aim of this study was to share an overview of the patterns and trends related to the prescription of antibiotics at a large, specialised children’s hospital in London, United Kingdom. A secondary objective of this study was to identify opportunities for improvement, thereby contributing to the advancement of AMS practices. 

## 2. Results

14 surveys were conducted between March 2016 and April 2021. The median number of inpatients during each survey was 207, with a range of 173 to 245. Furthermore, 37.4% of inpatients were admitted to medical wards, 8.9% to surgical wards, 10.7% to the Paediatric Intensive Care Unit (PICU), 11.3% to the Neonatal Intensive Care Unit (NICU), 16% to postnatal wards, and the remainder to short-stay units. Within this cohort, it was found that 32.5% were prescribed at least one antimicrobial during the survey, with a range of 20–44% (Kruskal–Wallis test: *p* = 0.448). A review of 1042 prescriptions of antibiotics and 182 of other antimicrobials showed that, on average, 67.2% of all prescriptions had either an end-date or a review date documented (range: 43.8–89.3%). Of the total prescriptions, 66.4% adhered to hospital guidelines; 15.4% followed the advice of either a paediatric infectious diseases physician or microbiologist; and 1.6% followed external hospital recommendations. However, 93 prescriptions (7.6%) were not adherent to established guidelines. There was no clinical guideline available for the remainder of the prescriptions (9.4%).

Almost all antibiotics (99.8%) were administered systemically, with intravenous administration being the most common route (64.7%), followed by oral (33.9%) and inhaled (1.1%) routes. 

Amoxicillin and clavulanate (217 prescriptions) and gentamicin (159 prescriptions) were the most frequently prescribed agents. As illustrated in Figure 1, the preponderance of prescriptions was comprised of antimicrobials within the Watch category. In the Access category, gentamicin was the predominant antibiotic, followed by penicillin G (70 prescriptions) and flucloxacillin (59 prescriptions). Amoxicillin and clavulanate, cefotaxime (74 prescriptions), and ceftriaxone (72 prescriptions) were the most frequent agents in the Watch category. Among amoxicillin and clavulanate prescriptions, lower respiratory tract infections were listed as the most common indication. Conversely, cefotaxime and ceftriaxone were primarily prescribed empirically to treat infections causing sepsis and central nervous system conditions. The proportions of Reserve antibiotics remained relatively stable across all surveys, ranging from 0% to 9.1%. Meropenem (30 prescriptions) and colistin (14 prescriptions) were the main agents in the Reserve category. Detailed information regarding the specific clinical indications for the prescriptions is provided in Appendix C.

Surgical prophylaxis constituted 107 prescriptions (30.4%), with amoxicillin and clavulanate (53, 15.1%), gentamicin (12, 3.4%), and cefuroxime (8, 2.3%) being the principal agents in use. Additionally, 72 prescriptions (20.5%) were earmarked for antifungal prophylaxis, with fluconazole accounting for the majority (66, 18.8%), followed by clotrimoxazole (2, 5.7%), oral nystatin (2, 5.7%), and amphotericin B (2, 5.7%). 

Seventeen prescriptions were identified for non-infectious indications, of which 15 were for erythromycin and 2 for rifampicin. Indications for that included delayed gastric emptying and itching.

It is noteworthy that a total of 93 prescriptions (7.6%) were labelled as non-adherent to the hospital guidelines. Specifically, amoxicillin and clavulanate were the most frequently prescribed agents not adhering to the guidelines, accounting for 36 instances (38.7% of non-adherent prescriptions), followed by cephalexin (8, 8.6%) and gentamicin (7, 7.5%). Additional information concerning the specific indications that did not align with the established guidelines is summarised in Appendix C.

## 3. Materials and Methods

### 3.1. Setting 

Evelina London Children’s Hospital (ELCH), part of Guys’ and St Thomas’ NHS Trust, is a 140-bed inner-city children’s hospital. It plays crucial roles as an acute care centre for children residing in South London, as a tertiary referral hospital for the south-east of England, and as a major training centre within the London School of Paediatrics. Annually, the paediatric emergency department at ELCH caters to approximately 24,000 children and young people. ELCH is a specialised facility for paediatric cardiac surgery and renal replacement therapy, which includes a paediatric transplant programme. Additionally, the hospital features a Level 3 NICU and a PICU with 20 beds. Collaborations with other tertiary medical centres in London are established to address paediatric oncology, neurosurgery, and trauma medicine, as these subspecialties are not yet available within ELCH. 

ELCH provides readily accessible resources for empiric antibiotic therapy (Appendix B), including guidelines published on the hospital intranet and a free smartphone application. There are no restrictions on prescribers for antimicrobials available on the hospital’s formulary. Moreover, all departments benefit from the expertise of dedicated paediatric pharmacists and infection prevention and control practitioners. Weekly handshake stewardship rounds have been integrated into the medical ward (with the exception of paediatric neurology and metabolic medicine), PICU, and NICU since the year 2016. During these rounds, all antimicrobial prescriptions of inpatients are reviewed by a team comprising a paediatric infectious diseases physician and fellow, a microbiologist, a specialised infectious diseases pharmacist, and an infection prevention and control practitioner. Informal feedback is provided directly following these reviews, after which the prescription can be changed as per the discretion of the prescriber.

### 3.2. Data Collection

Point Prevalence Surveys (PPS) at approximately three-month intervals from 2016 to April 2021 were conducted to monitor antimicrobial prescriptions throughout the hospital. During each survey, a specialised pharmacist extracted all inpatient antimicrobial prescriptions through the hospital’s pharmacy electronic prescribing system, categorising them according to the UK-adapted WHO AWaRe classification (Appendix A). The indication for each prescription was recorded, and the adherence to the hospital’s guidelines was assessed. While hospital clinicians were informed about the existence of AMS activities, prescribers were not alerted to the specific days on which the PPS were conducted. 

### 3.3. Data Analysis and Presentation

The utilization of antibiotics and antimicrobials was described by the percentage of admitted children receiving these treatments in each survey. Antibiotics were further delineated by the proportion within each AWaRe category (Access, Watch, Reserve). The trends of the most commonly prescribed individual antimicrobials are expressed as the proportion per respective group in the AWaRe category. The trends in proportions were analysed using independent-samples Kruskal–Wallis tests and differences in proportions with Chi-Squared tests. A 2-tailed *p*-value of <0.05 was considered statistically significant. All analyses were performed in IPM SPSS version 29.

## 4. Discussion

Measuring antimicrobial prescribing provides directions for stewardship interventions and for other quality improvement initiatives [7,9]. However, measuring antimicrobial prescribing in paediatric populations presents challenges due to the lack of standardisation in dosing schedules [1]. In this retrospective analysis of audit data spanning a five-year period, we examined 1224 antimicrobial prescriptions collected within a large inner-city children’s hospital in London, United Kingdom. 

Throughout all recorded surveys, the proportion of antimicrobials classified as Access fell below the WHO recommended target of 60%. Instead, the prevailing trend indicated a predilection for agents categorised under the Watch category, with amoxicillin and clavulanate, cefotaxime, and ceftriaxone emerging as the most frequently prescribed antimicrobials. The main targets for improvements after our analyses are outlined in Table 1.

Amoxicillin and clavulanate’s classification in the Watch category in the UK classification rather than Access in the WHO classification is due to the heightened risk of antibiotic-related diarrhoea, particularly from *Clostridium difficile*, and to avoid unnecessarily broad antimicrobial therapy [8]. Although detailed insights into prescribing behaviours are outside the scope of this study, it is likely that alternative agents within the Access class would have been reasonable options for patients prescribed amoxicillin and clavulanate. 

Lower respiratory tract infections were the most frequent indication (40.1%) for amoxicillin and clavulanate. Hospital guidelines recommend either PO amoxicillin or IV ampicillin as first-line therapies for community-acquired pneumonia, reserving IV amoxicillin and clavulanate for severe cases. However, the definition of severe illness is not further specified. For instance, amoxicillin and clavulanate were more commonly prescribed for patients with lower respiratory tract infections who were admitted to the medical wards compared to those in the PICU (63.5% vs. 36.5%, *p* < 0.001). Notwithstanding, *Streptococcus pneumoniae* is the most common bacterial species causing community-acquired pneumonia in children, followed by *Haemophilus influenzae* and *Moraxella catarrhalis* [10]. Amoxicillin has good antibacterial activity against *Streptococcus pneumoniae*, but resistance rates of both *Haemophilus influenzae* and *Moraxella catarrhalis* to amoxicillin are rising worldwide. Consequently, in other guidelines, amoxicillin and clavulanate are reserved for pneumonia cases that did not respond to amoxicillin. As information on preceding treatments was not available for our analysis, some of the patients with respiratory infections may have had pretreatment with amoxicillin before they were switched to amoxicillin and clavulanate. Antibiotic therapy is not indicated for uncomplicated bronchiolitis, caused by viruses [11]. However, antibiotics are commonly prescribed for infants with severe bronchiolitis in an attempt to empirically treat a potential bacterial superinfection [12]. Intra-abdominal infections accounted for 10.6% of all amoxicillin and clavulanate prescriptions. In these polymicrobial infections, adequate coverage is imminent against Gram-negative bacteria (including *Escherichia coli*), Gram-positive bacteria (including *Enterococci*), and, contingent on the patient’s age and severity of the infection, anaerobic bacteria. The triple therapy of amoxicillin, gentamicin, and metronidazole, all Access agents, provides the same coverage as amoxicillin, clavulanate, and gentamicin. 

6.0% of all amoxicillin and clavulanate prescriptions were for skin and soft tissue infections. With *Staphylococcus aureus* and *Streptococcus pyogenes* being the most frequent bacteria responsible for this entity, antistaphylococcal penicillins, such as oxacillin or flucloxacillin, are first-line therapy for these infections, which is part of the Access class. The poor palatability of oral flucloxacillin can be overcome by mixing it with a small amount of food or liquid [13]. 

Cefotaxime and ceftriaxone were the second most frequently prescribed antimicrobials within the Watch group, with (suspected) sepsis being the most frequent indication. Since signs of sepsis are not specific and rapid treatment initiation is associated with better outcomes, initiatives have been relayed to institute rapid empiric treatment in patients in whom sepsis is suspected [14,15]. Therefore, stewardship activities to reduce the number of treatment initiations for this indication are unlikely to be successful. Nevertheless, appropriate diagnostics are important in order to de-escalate antimicrobials and, ideally, to discontinue antimicrobials when an alternative diagnosis becomes more likely. For neonatal sepsis, biomarker-guided therapy was associated with an earlier discontinuation of antimicrobials [16,17]. 

The prevalence of prescriptions for agents classified within the Reserve class remained relatively stable throughout the surveys (0–9.1%). While no formulary restrictions were in place, the absence of a haematology-oncology or bone marrow transplant unit in our centre limited the need for carbapenems and anti-pseudomonal antibiotic use.

Of all prescriptions, almost a third (28.8%) served prophylactic purposes, spanning various medical indications, including long-term prophylaxis against urinary tract infections, *Pneumocystis jirovecii* prophylaxis, exacerbation prophylaxis in bronchiectasis, and prophylaxis against *Ureaplasma* spp., which is associated with bronchopulmonary dysplasia [18]. In particular, macrolides, notably azithromycin, which has a very long half-life, have been employed for the latter indications. For all prophylactic indications, frequent review of the indication and consideration of discontinuation if the risk has decreased are crucial. 

Surgical prophylaxis emerged as prevalent in our surveys (107 prescriptions, or 30.4% of all prophylactic indications), with amoxicillin and clavulanate, gentamicin, and cefuroxime as the most frequently prescribed agents for this purpose. Prophylaxis exceeding 24 h generally does not yield improved clinical outcomes and should therefore be avoided to mitigate adverse drug events and the development of antimicrobial resistance [19,20,21]. More insights on the challenges in capability, opportunity, and motivation to prescribe prophylaxis for longer than necessary would be helpful in improving prescribing behaviour. 

Ninety percent of prescriptions for fungal prophylaxis were for infants who were admitted to the Neonatal Intensive Care Unit (NICU). In a nationwide study in neonatal units in the UK, Ferreras-Antolin et al. found that antifungal medications were prescribed for prophylactic reasons in 79.6% of cases. Noteworthy, 44.7% had ≥3 risk factors for invasive candidiasis, while the prevalence of microbiologically proven invasive candidiasis was only 5.4% [22]. These findings prompt the need for a national paediatric antifungal stewardship programme to promote rational prescribing in the NICU population.

Intravenous administration constituted the predominant route for the majority of prescriptions (64.7%), reflecting a common practice among hospitalised patients. However, many antimicrobials with high bioavailability were prescribed via the intravenous route, including clindamycin (92.9%), metronidazole (83.8%), and cotrimoxazole (54.1%). Therefore, there is a need to direct effort towards transitioning to oral administration as soon as this is possible, given the oral route’s efficacy in treating numerous infections and its potential to minimise issues related to line access and associated costs [23]. 

Our findings align with those of other studies. Tribble et al. undertook a cross-sectional analysis across 32 children’s hospitals in the United States between 2016 and 2017 [24]. Furthermore, 35.0% of the 34,927 children hospitalised on the survey days were on at least one antibiotic, of which 25.9% showed areas for improvement, e.g., bug-drug mismatch, prolonged surgical prophylaxis, and overly broad empiric therapy. In a follow-up study, Diggs et al. conducted PPS in 28 US children’s hospitals between 2019 and 2020. Moreover, 13.8% out of a total of 13,344 antibiotic orders were deemed inappropriate. The highest number of inappropriate orders were in the paediatric intensive care units and medical units, while prolonged or unnecessary surgical prophylaxis was the most common inappropriate order for surgical inpatients. Furthermore, an ID consultation in the preceding 7 days was associated with a lower likelihood of inappropriate antibiotic orders. Wang et al. conducted quarterly PPS in 16 Chinese general and children’s hospitals in 2019 [25]. Moreover, 66.1% of the cohort, encompassing 22,327 hospitalised children, received at least one antibiotic prescription. Lower respiratory tract infections (LRTI) were the most prevalent reason for antibiotic prescription (43.2%), and 70.4% of all antibiotics were agents of the WHO Watch class, mostly third-generation cephalosporins and penicillins/β-lactamase inhibitors, macrolides, and carbapenems. 

Our study has several limitations. First, the retrospective nature of the analysis did not allow for detailed information on the rationale behind prescribing decisions. Hence, our study lacks the ability to identify reasons for prescriptions that deviated from guidelines. Future research should emphasise the exploration of factors underlying non-adherence to guidelines, as these could serve as potential targets for stewardship initiatives. Second, constraints on stewardship resources hindered data collection between April 2016 and September 2017. Despite identifying consistent patterns through the surveys, data beyond the survey dates, especially for antimicrobials prescribed only on specific weekdays, was overlooked. This limitation stems from the inherent nature of PPS data collection, as our centre lacked an electronic system for continuous measurements of antimicrobial use during the study period. Nonetheless, since prescribers were unaware of the survey days, we do not expect this to significantly impact our data. Third, data for this study have been partially collected during the COVID-19 pandemic, a period characterised by a decrease in elective admissions and a decline in the occurrence of bacterial infections circulating in the community [26,27]. These factors had an impact on the patterns of antibiotic prescribing. 

This study identifies targets for improvement in antimicrobial prescribing. If achieving a higher proportion of antibiotics in the Access category is desired, ongoing efforts will be needed to replace amoxicillin and clavulanate with antimicrobials in the Access category, and there should be a focus on targeting therapy (ideally narrow-spectrum agents, if effective) as soon as possible once new information on the diagnosis of sepsis becomes available. Guidelines on indications for both medical and surgical prophylaxis need to be better implemented. Furthermore, our study highlights that ongoing stewardship activities are essential since stewardship interventions are known to have a limited time span [28,29]. This is particularly true in teaching hospitals, such as our centre, with a high turnover of prescribers. Primhak et al. recently reported their success in achieving significant and sustained improvement in adherence to antimicrobial guidelines in a New Zealand Children’s Hospital following the introduction of a prescribing support app [30]. Our centre had an antibiotic prescribing smartphone application during the study period. Further research is required to pinpoint areas for enhancement that can provide better clinical decision support tools to prescribers. 

## 5. Conclusions

Over a 5-year period, an average of 32.5% of all inpatients were prescribed at least one antimicrobial, a trend that remained consistent across all surveys (range: 20–44%, *p* = 0.448). The majority of prescriptions fell within the Watch class, with amoxicillin, clavulanate, and cephalosporins emerging as the most frequently prescribed agents. Noteworthy areas for improvement include reducing amoxicillin and clavulanate prescriptions for lower respiratory tract infections, enforcing review dates for patients treated with third-generation cephalosporins for suspected sepsis, and reviewing the indications for prophylactic antimicrobials. Adherence with hospital guidelines, while generally positive, also revealed areas for improvement, particularly with amoxicillin and clavulanate being frequently prescribed in non-adherent instances. Ongoing stewardship activities are crucial to sustaining the effects of stewardship efforts and improving the quality of medical care. 

## Figures and Tables

**Figure 1 antibiotics-13-00172-f001:**
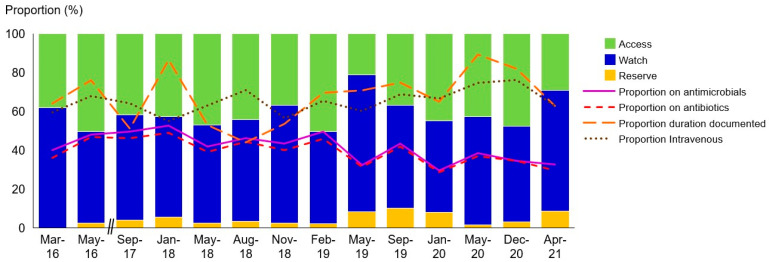
Proportion of antibiotic prescriptions per AWaRe category (stacked bar charts) and patients on antimicrobials and antibiotics during the survey. A total of 352 prescriptions (28.8% of antimicrobial prescriptions) had a prophylactic indication listed, with 147 allocated (41.8%) for long-term medical prophylaxis. Azithromycin (38, 10.8%) and cotrimoxazole (27, 7.7%) were the most frequently prescribed agents for this purpose. Additional agents employed for medical prophylaxis included trimethoprim (24, 6.8%), penicillin G (15, 4.3%), cephalexin (13, 3.7%), colistin (8, 2.3%), amoxicillin and clavulanate (5, 1.4%), valganciclovir (3, 8.5%), and acyclovir (1, 0.3%).

**Table 1 antibiotics-13-00172-t001:** Improvement targets following the PPS.

Antimicrobial Agent	Category	Target for Improvement
Co-trimoxazole	Access	33/37 prescriptions served prophylactic indications (both medical and surgical). Therefore, frequent revisions of indications for long-term prophylaxis could potentially decrease the number of prophylactic prescriptions.
Amoxicillin and clavulanate	Watch	87/217 prescriptions had lower respiratory tract infection labelled as an indication, which is the first line agent for severe community-acquired pneumonia and hospital-acquired pneumonia. Specific criteria to classify cases as severe may increase the prescription of the first-line agent, amoxicillin. Intra-abdominal infections were labelled as an indication for 23/217 prescriptions. The combination of amoxicillin, gentamicin, and metronidazole (all Access agents) results in similar coverage.
Azithromycin	Watch	41/43 prescriptions were labelled with a prophylactic indication. Restricting indications and establishing scheduled revision dates for long-term prophylactic use has the potential to decrease the number of prescriptions.
Cefotaxime	Watch	Among the 74 prescriptions, 63 were labelled with sepsis as an indication. However, microbiological findings guided this indication for only seven prescriptions. Reconsideration of the working diagnosis of sepsis is advisable if the blood culture remains negative, potentially allowing for earlier treatment discontinuation. Additionally, for infants older than 28 days who are not on parenteral nutrition or on any calcium-containing infusion fluids, cefotaxime can be replaced by ceftriaxone, offering the advantage of easier administration (once daily versus four times per day).
Ciprofloxacin	Watch	The use of fluoroquinolones should be restricted, given the very low barrier to resistance. Therefore, it is not a standard agent in our formulary, but it is considered a case-by-case scenario if it is the only available agent via the oral route.
Meropenem	Reserve	Carbapenems serve as last-resort agents; nevertheless, microbiological findings guided only 10 out of 30 meropenem prescriptions in this study. Introducing scheduled review dates following meropenem prescriptions could provide an opportunity to evaluate and consider alternative agents.
Fluconazole	Antifungals	54/76 prescriptions provide prophylactic indications, in particular in the neonatal population. Scheduled revision dates for long-term prophylactic use can reduce the number of prescriptions.

## Data Availability

Data are available upon reasonable request from the corresponding author.

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
