# Peer review of "Trends in Antibiotic Use in a Large Children’s Hospital in London (United Kingdom): 5 Years of Point Prevalence Surveys"

_antibiotics, 2024, doi:10.3390/antibiotics13020172_

Round 1

Reviewer 1 Report

Comments and Suggestions for Authors

l  It is recommended that the results part be moved after the methodology part.

l  Give the table 1 is too big, it may be better to put the detail list of each agent and the corresponding indication to attachments and have bit more description in the paragraph.

l  Why did the study do 14 surveysHow to choose the date? Why most interval was about 3 months, while some was 4, 5 months and even 7 months.

l  It is described in line 107 that It is noteworthy that a total of 93 prescriptions (7.6%) were labeled as non-adherent with the hospital guidelines. Have the reasons be investigated?

l  In discussion part, it is better to divide several points and then give evidences. For the situation of prescription of antibiotics, what are the special meaning for children, is there some differences between children and adults?

Author Response

Dear reviewer 1,

Thank you sincerely for taking the time to review our manuscript. Your insightful comments and constructive feedback have been invaluable in enhancing the overall quality of our work. We appreciate your thorough evaluation. Please find our point by point responses below. Furthermore, we refer to the revised version of our manuscript.

It is recommended that the results part be moved after the methodology part.

Thanks for this suggestion. Yet, the standard Antibiotics MDPI style defines that the Results section should precede the Methods.

Give the table 1 is too big, it may be better to put the detail list of each agent and the corresponding indication to attachments and have bit more description in the paragraph.

In line with this feedback and that of other reviewers, we moved this table to the Appendix file. Instead, we inserted a table outlining the main targets for improvement in the discussion section (table 1).

Why did the study do 14 surveys?How to choose the date? Why most interval was about 3 months, while some was 4, 5 months and even 7 months.

The audits are scheduled quarterly on a routine basis, however conducting them at this interval was significantly constrained by hospital resources allocated to antimicrobial stewardship. As outlined in lines 284-286, there have no audits been performed between April 2016 and September 2017. This is also depicted in figure 1 (the -//- between May 2016 and Sept 2017).

It is described in line 107 that “It is noteworthy that a total of 93 prescriptions (7.6%) were labeled as non-adherent with the hospital guidelines”. Have the reasons be investigated?

Thank you for raising this comment. In line with the feedback of reviewer 3, we added data on the frequency of prescriptions that were not adherent to guidelines (Appendix 3). Yet, as this study represents an analysis of retrospectively collected audit data, we do not have detailed insight into the reasons for prescribing, since we did not clarify that with the prescribers. Future research should address factors related to non-adherence to guidelines, as these may be a target for further stewardship activities. We outline on this in the discussion section.

In discussion part, it is better to divide several points and then give evidences. For the situation of prescription of antibiotics, what are the special meaning for children, is there some differences between children and adults?

Thanks for this comment. We start the discussion section with a summary of our findings, and subsequently reflect on general trends in prescribing throughout the surveys, indications for prescribing and on targets for improvement. Last, we compare our work with existing literature on paediatric antimicrobial use, which is indispensible given the limited literature available on this topic. 

Reviewer 2 Report

Comments and Suggestions for Authors

Dear Authors,

I've carefully read your manuscript and I have these comments and suggestions for you:

1.      No point after the title

2.      Detail what Co-amoxiclav is, since this product, under this commercial name, is not available in all countries. You should use the names of the substances, amoxicillin+potassium clavulanate

3.      The space between the lines in some manuscript parts is bigger than in other

4.      The Introduction part is too general, I suggest you detail and improve

5.      What do PICU and NICU in line 59 stand for?

6.      I suggest you detail what the “Watch category “ is

7.      The title of Table 1 should appear before the table itself

8.      I suggest you prepare a figure with the most used antibiotics, the most incriminated in causing adverse reactions, etc. The organization of data into figures and tables make the reading of the manuscript easier and more interesting for the readers

9.      I consider a good aspect of your paper the presentation of the study’s limitations

TThe letter size in Conclusion seems to be smaller than in other manuscript part

TThe references are well chosen and in agreement with the research theme

Author Response

We would like to express our gratitude for your thoughtful review of our manuscript. Your detailed comments and suggestions have been incredibly helpful in refining our work. Please find the point-by-point responses to your comments below.

No point after the title

Thanks for this suggestion- we amended that accordingly.

Detail what Co-amoxiclav is, since this product, under this commercial name, is not available in all countries. You should use the names of the substances, amoxicillin+potassium clavulanate

Yes, that may be confusing. We substituted co-amoxiclav by “amoxicillin and clavulanate” throughout the manuscript, as using ‘amoxicillin+potassium clavulanate’ would be excessively long.

The space between the lines in some manuscript parts is bigger than in other

Thanks for flagging this. We reviewed the manuscript, and adjusted the line spacing across the manuscript.

The Introduction part is too general, I suggest you detail and improve

We appreciate your comment. We amended parts of the introduction, and (in line with point 6), we defined the different AWaRe categories.

What do PICU and NICU in line 59 stand for?

We specified these abbreviations in the revised manuscript.

I suggest you detail what the “Watch category “ is

Please refer to point 4.

The title of Table 1 should appear before the table itself

We agree, and moved table 1 to the Appendix.

I suggest you prepare a figure with the most used antibiotics, the most incriminated in causing adverse reactions, etc. The organization of data into figures and tables make the reading of the manuscript easier and more interesting for the readers

Thanks for this suggestion. We moved table 1 to the Appendix, and added a table with the main findings and targets for improvement.

I consider a good aspect of your paper the presentation of the study’s limitations

Thanks so much for this feedback.

The letter size in Conclusion seems to be smaller than in other manuscript part

The font size has been amended in the revised manuscript.

The references are well chosen and in agreement with the research theme

Thanks so much for this remark.

Reviewer 3 Report

Comments and Suggestions for Authors

Thanks for giving me an opportunity to review. Here are some suggestions:

- Line 80: the word "to" is missing.

- Line 129: Smartphone is misspelt

Page 6: Metronidazole "Eear" is misspelt

- What was the age of the children treated? I understand you wanted a simple descriptive study but by clubbing all the years the trend can't be observed. Can you perform a time series analysis.

- Can you provide details on the Table spanning Page 6 to Page 13 of the PDF on how many of these antibiotics were properly prescribed. How many had a culture prior to administration (especially Watch or Reserve)?

Author Response

Thank you for your diligent review of our manuscript. Your comments have played a crucial role in shaping the final version of our work. We are grateful for your attention to detail and the constructive nature of your feedback. Please find our point-by-point responses to your feedback below.

Line 80: the word "to" is missing.

Line 129: Smartphone is misspelt

Page 6: Metronidazole "Eear" is misspelt

Thanks so much for reading our manuscript so diligently. We changed these spelling errors accordingly.

What was the age of the children treated? I understand you wanted a simple descriptive study but by clubbing all the years the trend can't be observed. Can you perform a time series analysis.

As we report the cumulative data obtained via routine audits, identifiers of the patients to whom the antibiotics were prescribed were not recorded during the audits. However, the main aim of our study was to identify trends in antimicrobial prescribing and identifying targets for improvement- for which demographic data at the patient level would be ideal, but not necessary. 

Can you provide details on the Table spanning Page 6 to Page 13 of the PDF on how many of these antibiotics were properly prescribed. How many had a culture prior to administration (especially Watch or Reserve)?

As per the feedback of reviewer 1 and 2, we moved the table to the Appendix and extended the table with frequencies of adherence to guidelines, the number of prescriptions that were guided by microbiology findings, and the number of prescriptions that were directed by external centers.

Round 2

Reviewer 2 Report

Comments and Suggestions for Authors

Dear Authors,

Thank you for considering my comments and suggestions and for making the changes into your manuscript.